# A Study on Design of S-Duct Structures and Air Intake for Small Aircraft Applied to High Strength Carbon–Epoxy Composite Materials

**DOI:** 10.3390/ma15093001

**Published:** 2022-04-20

**Authors:** Semyeong Lim, Won Choi, Hyunbum Park

**Affiliations:** 1Department of Mechanical Engineering, Kunsan National University, 558 Daehak-ro, Miryong-dong, Gunsan 54150, Korea; amuse3030@daum.net; 2Aero Machinery R&D Center, Hanhwa Aerospace, Asanvalleynam-ro, Dunpo-Myeon, Asan 31409, Korea; tailer49@daum.net

**Keywords:** unmanned aerial vehicles, air intakes, structural design, finite element analysis, carbon–epoxy composite

## Abstract

Recently, many structural parts using composite materials are being applied to small aircraft and UAV in the world. The aim of this work is to design the engine intake structure of a small aircraft. For structural safety evaluation, a finite element analysis method was applied. In this work, structural design and numerical analysis of air intake and s-duct structures for small aircraft were performed. The target structure is composed of an s-duct and a cylindrical intake structure. Firstly, an investigation of the mechanical properties of carbon/epoxy material was conducted. The distributed pressure load and acceleration condition was applied to the structural design. The structural design load was investigated considering safety factors. The structural analysis was performed to analyze the validity of the design results. Through the structural analysis using the finite element analysis method, it was confirmed that the designed air intake structure is safe. The manufacturing of the prototype structure will be carried out based on the designed result.

## 1. Introduction

Applications of composite materials have been diversified for lightweight aircraft designs. This is because composite materials are very advantageous in weight lightening of the existing metallic material of the aircraft. Carbon fiber composite materials are mostly applied to design aircraft structures. In this study, carbon/epoxy composite materials were applied to carry out structural design and analysis research for the air intake structures where air is flowed inside the engine.

As a result of reviewing the preceding research results in this study, it was determined that research on aircraft air intake component design has been significantly conducted. However, most previous studies were carried out on the aerodynamic shape design component. There were insufficient research results that applied composite materials to conduct lightweight design in terms of structural design. Therefore, carbon/epoxy composite materials were applied to carry out the lightweight design and analysis research for the engine intake in this work.

As a result of investigation on previous study results related to aircraft intake design and analysis, Pedro David Bravo-Mosquera et al. conducted integration assessment of conceptual design and intake aerodynamics of a non-conventional air-to-ground fighter aircraft. These studies provided extensive data to evaluate the effects of specific aircraft design variables on dorsal intake performance, such as the integration of the cockpit, the boundary layer diverter, and several delta wing planforms [1].

Vittorio Vercillo et al. performed a study on modeling of the icing of air intake protection grids of aircraft engines. In this work, a physical model was validated for icing of air intake protection grids of engines [2].

F. Bagnoli et al. conducted research on failure analysis of an aircraft auxiliary power unit air intake door. In this work, the design of a new door as well as the material were introduced, thus obtaining the necessary functional improvements [3].

Zeyang Zhou et al. carried out a study on the mixed design of radar/infrared stealth for an advanced fighter intake and exhaust system. To improve the overall stealth performance of the aircraft’s intake and exhaust systems, a mixed design approach (MDA) is presented in this work [4].

Mohammad R. Soltani el al. performed a study on numerical simulation and parametric study of a supersonic intake. In this work, a computational fluid dynamics code was developed to compute the flow inside and around a supersonic external compression axisymmetric intake [5].

Wenbiao Gan et al. conducted a study on design optimization of a three-dimensional diffusing S-duct using a modified SST turbulent model. This paper examines design optimization of a three-dimensional diffusing S-duct using a modified k–ω shear stress transport (SST) turbulent model as a turbulence prediction method [6].

H. Kim et al. performed research on shape design optimization of embedded engine inlets for the N2B hybrid wing–body configuration. In this work, the N2B hybrid wing–body aircraft with embedded engines was conceptually designed to meet environmental and performance goals for the generation transport set [7].

Guoping Huang et al. conducted a study on the design method of the internal waverider inlet under the non-uniform upstream for inlet/forebody integration. In this work, a novel bump-integrated three-dimensional internal waverider inlet (IWI) design method is presented for high-speed inlet/forebody integration [8].

Lee M. et al. performed research on the conceptual redesign of the B-1B bomber inlets for improved supersonic performance. This paper presents a conceptual study of two alternative inlet concepts for the United States Air Force B-1B bomber to provide improved supersonic performance with the expansion of capabilities for high-altitude, high-speed flight at Mach 2.0 [9].

Shuvayan Brahmachary et al. conducted a study on multi-point design optimization of a high-performance intake for scramjet-powered ascent flight. This paper presents the results and insights obtained from a first-ever multi-point multi-objective optimization study of an axisymmetric scramjet intake conducted by means of surrogate-assisted evolutionary algorithms coupled with high-fidelity computational fluid dynamics [10].

Guoxing Song et al. performed a study on experimental simulation methodology and spatial transition of complex distortion fields in a S-shaped inlet. In this paper, a subsonic S-shaped inlet test rig was constructed, and eight configurations of curved-edge plate-type distortion generators were installed in the intake channel to create a complex distorted flow field downstream [11].

Michael M. Wojewidka et al. carried out a numerical study of complex flow physics and coherent structures of the flow through a convoluted duct. In this work, phase analysis of modes in convoluted ducts are reported in the public domain for the first time and offer promise of a tailored, active flow control strategy [12].

Liu Jun et al. performed investigation of translation scheme turbine-based combined-cycle inlet mode transition. The steady and unsteady flow characteristics of the inlet mode transition were studied through wind tunnel tests and numerical simulations [13].

Gyuho Kim et al. conducted a study on failure analysis of an aircraft APU exhaust duct flange due to low cycle fatigue at high temperatures. A detailed investigation of crack-induced fracture surface was conducted using scanning electron microscopy (SEM) and computer-aided thermal-stress analysis [14].

Miroslaw Wroblewski et al. conducted a study on areas of investigation into air intake systems for the impact on compressor performance stability in aircraft turbine engines. This paper presents selected areas of research into the surge and stall of axial compressors used in aircraft turbine engines based on scientific publications in recent years [15].

A. Kozakiewicz et al. performed a study on the impact of the intake vortex on the stability of the turbine jet engine intake system. The article presents a numerical analysis of the intake system of a turbine jet engine in terms of parameter stability along its duct, following the occurrence of an intake vortex [16].

A. Kozakiewicz et al. conducted an analysis of the gust impact on the inlet vortex formation of the fuselage-shielded inlet of a jet-engine-powered aircraft. In this study, the analysis of the impact of changes in speed, angle, and the direction of gust on vortex development was conducted [17].

C. Soutis performed a study on fiber-reinforced composites in aircraft construction. In this paper, a review of recent advances using composites in modern aircraft construction is presented and it is argued that fiber-reinforced polymers, especially carbon-fiber-reinforced plastics (CFRP) can and will in the future contribute more than 50% of the structural mass of an aircraft [18].

A. Grbovic et al. carried out a study on the experimental and numerical evaluation of fracture characteristics of composite material used in aircraft engine cover manufacturing. In this work, the design of the composite engine cover of light aircraft was roughly presented [19].

M. Finley performed a study on composites that provide greener aircraft engines. In this work, the trend of composite materials used in aircraft engines was presented [20].

Many research works on aerodynamic design for air intake of aircraft were performed. However, little research work has been carried out to design and conduct a numerical analysis of small aircraft structures using composite materials. The aim of this work is to design the engine intake structure of a small aircraft. In this study, a high-strength carbon/epoxy composite was applied to conduct the structural design and analysis of small aircraft structures. For structural safety evaluation, a finite element analysis method was applied. Finally, structural design and numerical analysis of air intake and s-duct structures for small aircraft were performed.

## 2. Mechanical Properties of Carbon/Epoxy Materials

In this study, a high-strength carbon/epoxy composite was adopted for structural design. The carbon fiber and epoxy matrix is a material optimized for aircraft structures to reduce weight. Recently, most aircraft are designed by adopting carbon fiber for weight reduction. Therefore, carbon fiber was adopted for the structural design of the target small aircraft. In this work, investigation of mechanical properties of a carbon/epoxy composite is performed as a precedent study on the design of air intake structures using a carbon fiber composite. The carbon fiber/epoxy composite specimens were manufactured by the autoclave manufacturing process. The specimen test was performed by the ASTM strength test method. The tensile strength test was performed by ASTM D3039 [21]; the compressive strength test was performed by ASTM D6641 [22]; the flexural strength test was performed by ASTM D790 [23]; and the shear strength test was performed by ASTM D5379 [24]. The Universal Testing Machine of Shimadzu AG-250KNX was applied. The equipment was manufactured by SUNGSAH HI TECH in the Republic of Korea. The test speed of the equipment was 0.001~500 mm/min. Figure 1, Figure 2, Figure 3 and Figure 4 show the tested result of a tensile and compression specimen [25]. The mechanical properties after the specimen test were used for the structural design. The design using carbon/epoxy composite was performed after investigation of the mechanical properties of the specimen. In this work, the fracture shape was also investigated and analyzed by applying the failure theory. The Tsai-Wu failure criterion was adopted [26,27]. Figure 2 and Figure 4 show the fiber failure shape of the carbon composite specimen. In the case of tensile specimens, the fiber fracture behavior after matrix damage is shown. In the case of compression specimens, the fibers at both ends are broken by compression, showing the type of damage.

## 3. Structural Design

This study carried out structural design and analysis on the s-duct and air intake in the front of a small aircraft engine. The materials applied for structural design were carbon/epoxy composite ones. Through the preceding research, as the material was applied to the manufacture of the small aircraft, mechanical properties of F6273C-07M, carbon/epoxy fabric prepreg of Toray Industries Inc., were applied. The mechanical properties from the specimen test were applied to the design. For the target structure’s thickness, the thickness of one ply of prepreg was 0.22 mm, and accordingly, the final thickness and stacking angle were determined.

The design loading conditions are the pressure distribution load and acceleration status. Table 1 shows the load conditions for structural design. The structural design was carried out considering the distributed load and acceleration conditions. Figure 5 shows pressure distribution by CFD analysis result. The safety factor 1.5 was considered. The design requirements were applied according to the composite materials handbook (CMH-17-3G) [28].

Firstly, the design method of the netting rule was used for the initial design. The design method for determining the mixture was used to detail the design. The target structure consists of two parts: a plate part and a curved panel. The design of the plate part was performed considering the laminate constitutive theory [29,30]. The curved panel was designed considering the curved beam theory [31,32]. Since a curved beam is such a generic feature, we treated the specific configuration for which transverse tensile stresses become significant and may precipitate the failure of a structure. We assumed that characterization as a panel stress elastic problem is adequate. Figure 6 shows the curved beam configuration. A solution to the posed curved beam problem results from a formulation in polar coordinates and the identification of an airy-type stress function. We used the equation of the radial and tangential stress component as follows. The following equations were used to design the curved structure of the duct; *σ_r_* is radial stress, *k* is shear coefficients, and *σ_θ_* is tangential stress. The structural design was performed by comparing the stress calculated using the following equation and the yield strength of the material under the applied load:(1)σr=−Mb2hG1−1−abk+11−ab2krbk−1⋯−1−abk−11−ab2kark+1
where,
(2)$$k=\frac{E_{\theta }}{E_{r}}\;$$
(3)G=121−ab2 ⋯−k1−ab2k1−abk+12k+1−1−abk−12k−1ab2
(4)σθ=−Mb2hG1−1−abk+11−ab2kkrbk−1⋯+1−abk−11−ab2kkark+1

A similar expression was derived by Timoshenko for an isotropic curved beam. In this study, the design of a curved structure was performed using the above equation. The geometric factors (*b*, *a*, *h,* and *r*) are illustrated in Figure 6. Finally, the thickness of the target structure was defined. The design result of the initial structure was determined to be 1.1 mm. The laminate sequence is [45°/0°/45°/0°/45°]. The thickness of 1 ply is 0.22 mm.

## 4. Structural Analysis of S-Duct and Results Discussion

Structural analysis was conducted for the structural design result to examine structural safety. The commercial software used was MSC. Nastran, which is a finite element analysis method, was used to carry out the structural analysis. The structural analysis was carried out to analyze stress, displacement, and buckling.

A three-dimensional shape for structural analysis was investigated to generate a finite element analysis model. The result of generating the finite element analysis model to carry out structural analysis in this study is shown in Figure 7. The total number of elements generated for structural analysis was 2,161,305 elements. The four-node composite shell element was applied for the finite element model. For the boundary condition, the connecting part was applied as the fixed boundary condition. Figure 8 shows the boundary condition for structural analysis. For the application load, the pressure distribution load and acceleration condition were taken into account to carry out structural analysis by applying the safety factor of 1.5 for each case.

The load case one is a case of −6G acceleration in the direction of *z*. As a result of structural analysis of the load case one, the maximum stress was examined as 2.62 MPa tension and 1.80 MPa compression at layer one which is the external outer-most layer. It was examined that internal layer two had 1.72 MPa tension, and 1.09 MPa compression, and layer five, which is the internal outer-most layer, had 1.93 MPa tension and 2.51 MPa compression. Therefore, it was confirmed to be safe. The result of displacement analysis was investigated as 0.10 mm at the side part. The displacement was confirmed to be safe. As for the buckling analysis, the load factor of first buckling was 35. Therefore, it was examined to be a sufficiently stable structure against buckling. Figure 9, Figure 10 and Figure 11 show the result of stress and displacement through the structural analysis of load case one.

The load case two is a case of +3G acceleration in the direction of *z*. This case was confirmed to be the same result as the load case one. As a result of structural analysis, the maximum stress was examined as 2.62 MPa tension and 1.80 MPa compression at layer one, which is the external outer-most layer. It was examined that internal layer two had 1.72 MPa tension and 1.90 MPa compression, and layer five, which is the internal outer-most layer, had 1.93 MPa tension and 2.51 MPa compression; therefore, it was confirmed to be safe enough. The displacement analysis result was examined as 0.10 mm at the side part, so the displacement was also confirmed to be safe enough. As a result of the buckling analysis, the first buckling load factor was 35, so it was examined to be a sufficiently stable structure against buckling.

Load case three is a case of +2G acceleration in the direction of *y*. For this case, as a result of structural analysis, the maximum stress was examined as 1.90 MPa tension and 1.07 MPa compression at layer one which is the external outer-most layer. It was examined that internal layer two had 8.17 MPa tension and 9.65 MPa compression, and layer five, which is the internal outer-most layer, had 11.7 MPa tension and 15.8 MPa compression, so it was confirmed to be safe enough. The displacement analysis result was examined as 2.43 mm at the internal part. As a result of the buckling analysis, the first buckling load factor was 2.82. Therefore, it was examined to be a sufficiently stable structure against buckling.

The load case four is a case of +20G acceleration in the direction of *x*. As a result of structural analysis, the maximum stress was examined as 5.28 MPa tension and 2.36 MPa compression at layer one, which is the external outer-most layer. It was examined that internal layer two had 2.79 MPa tension and 1.74 MPa compression, and layer five, which is the internal outer-most layer, had 2.75 MPa tension and 3.57 MPa compression; therefore, it was confirmed to be safe enough. The displacement analysis result was examined as 0.18 mm at the side part. As a result of the buckling analysis, the first buckling load factor was 2.8. Therefore, it was examined to be a sufficiently stable structure against buckling.

The load case five is a case of +20G acceleration in the direction of *y*. This case was also confirmed to be the same as the load case three. As a result of structural analysis, the maximum stress was examined as 1.90 MPa tension and 1.07 MPa compression at layer one, which is the external outer-most layer. It was examined that internal layer two had 8.17 MPa tension and 9.65 MPa compression, and layer five, which is the internal outer-most layer, had 11.7 MPa tension and 15.8 MPa compression; therefore, it was confirmed to be safe enough. The displacement analysis result was examined as 2.43 mm at the internal part. As a result of the buckling analysis, the first buckling load factor was 2.8. Therefore, it was examined to be a sufficiently stable structure against buckling.

As the load case six is a case of +20G acceleration in the direction of *z*, it was confirmed to be the same as load cases one and two. As a result of the structural analysis, the maximum stress was examined as 2.62 MPa tension and 1.80 MPa compression at layer one, which is the external outer-most layer. It was examined that the internal layer two had 1.72 MPa tension and 1.90 MPa compression, and layer five, which is the internal outer-most layer, had 1.93 MPa tension and 2.51 MPa compression; therefore, it was confirmed to be safe enough. The displacement analysis result was examined as 0.10 mm at the side part, so the displacement was also confirmed to be safe enough. As a result of buckling analysis, the first buckling load factor was 35, so it was examined to be a sufficiently stable structure against buckling.

In this study, structural analysis of the designed result was performed. As a result of examining the stress and displacement analysis results, it was confirmed to be a safe structure. Table 2 shows a summary of the structural analysis results of the s-duct structure. The structural analysis was performed for a total of six cases in this study. The tensile stress and compressive stress were confirmed to be sufficiently safe as a result of examining the safety factor. It was confirmed that it was sufficiently safe when compared with the results of other studies on the design and analysis of aircraft to which the composite material was applied.

## 5. Structural Analysis of Air Intake and Results Discussion

After performing the structural analysis of the s-duct component, the structural analysis of the air intake was performed. The three-dimensional shape of the structural design result was analyzed to generate a finite element analysis model. The result of generating the finite element analysis model to carry out structural analysis in this study is shown in Figure 12. The total number of elements generated for structural analysis was 34,167. For the boundary condition, the connecting part was applied as the fixed boundary condition. Figure 12 shows the boundary conditions. For the application load, the pressure distribution load and acceleration condition were taken into account to carry out structural analysis by applying the safety factor of 1.5 for each case. In this work, the contents of previous studies were improved and applied [33].

Load case 1 is a case of −6G acceleration in the direction of *z*. As a result of the structural analysis of load case one, the maximum stress was examined as 1.27 MPa tension and 1.80 MPa compression at layer one, which is the external outer-most layer. It was examined that internal layer two had 0.486 MPa tension and 1.09 MPa compression, and layer five, which is the internal outer-most layer, had 1.17 MPa tension and 1.27 MPa compression. Therefore, it was confirmed to be safe enough. The displacement analysis result was examined as 3.94 mm at the connecting part, so the displacement was also confirmed to be safe enough. As a result of buckling analysis, the first buckling load factor was 1.1. Therefore, it was examined to be a sufficiently stable structure against buckling.

Load case two is a case of +3G acceleration in the direction of *z*. This case was confirmed to be the same result as load case one. As a result of the structural analysis, the maximum stress was examined as 1.27 MPa tension and 1.80 MPa compression at layer one which is the external outer-most layer. It was examined that internal layer two had 0.486 MPa tension and 1.09 MPa compression, and layer five, which is the internal outer-most layer, had 1.17 MPa tension and 1.27 MPa compression; therefore, it was confirmed to be safe enough. The displacement analysis result was examined as 3.94 mm at the connecting part, so the displacement was also confirmed to be safe enough. As a result of the buckling analysis, the first buckling load factor was 1.1, so it was examined to be a sufficiently stable structure against buckling.

Load case three is a case of +2G acceleration in the direction of *y*. For this case, as a result of the structural analysis, the maximum stress was examined as 2.0 MPa tension and 2.22 MPa compression at layer one, which is the external outer-most layer. It was examined that internal layer two had 0.845 MPa tension and 1.74 MPa compression, and layer five, which is the internal outer-most layer, had 1.21 MPa tension and 2.15 MPa compression; therefore, it was confirmed to be safe enough. The displacement analysis result was examined as 46 mm at the side. As a result of the buckling analysis, the primary buckling load factor was 0.038. Therefore, it was examined to be a little bit unstable locally against displacement and buckling.

The load case four is a case of +20G acceleration in the direction of *x*. As a result of the structural analysis, the maximum stress was examined as 7.37 MPa tension and 19.6 MPa compression at layer one, which is the external outer-most layer. It was examined that internal layer two had 4.86 MPa tension and 20.9 MPa compression, and layer five, which is the internal outer-most layer, had 9.10 MPa tension and 11.0 MPa compression; therefore, it was confirmed to be safe enough. The displacement analysis result was examined as 46.6 mm at the top, so the displacement was considered to be a little bit high locally. As a result of buckling analysis, the first buckling load factor was 0.1, so it was examined to be a locally unstable structure against buckling.

Load case five is a case of +20G acceleration in the direction of *y*. This case was also confirmed to be the same as load case three. As a result of the structural analysis, the maximum stress was examined as 2.0 MPa tension and 2.22 MPa compression at layer one, which is the external outer-most layer. It was examined that internal layer two had 0.845 MPa tension and 1.74 MPa compression, and layer five, which is the internal outer-most layer, had 1.21 MPa tension and 2.15 MPa compression; therefore, it was confirmed to be safe enough. The displacement analysis result was examined as 46 mm at the side. As a result of the buckling analysis, the first buckling load factor was 0.038, so it was examined to be a little bit unstable locally against displacement and buckling.

As load case six is a case of +20G acceleration in the direction of *z*, it was confirmed to be the same as load cases one and two. As a result of the structural analysis, the maximum stress was examined as 1.27 MPa tension and 1.80 MPa compression at layer one, which is the external outer-most layer. It was examined that internal layer two had 0.486 MPa tension and 1.09 MPa compression, and layer five, which is the internal outer-most layer, had 1.17 MPa tension and 1.27 MPa compression; therefore, so it was confirmed to be safe enough. The displacement analysis result was examined as 3.94 mm at the connecting part, so the displacement was also confirmed to be safe enough. As a result of the buckling analysis, the first buckling load factor was 1.1, so it was examined to be a sufficiently stable structure against buckling.

As a result of the structural analysis on the structural design result of the aircraft component with composite materials applied in this study, it was examined that the plate and cylinder structure shape, which is the connecting part, was vulnerable to buckling. Therefore, the final design result was that the number of the relevant part’s laminated layers was increased to redesign, and as a result of the final structural analysis, it was confirmed to secure structural safety. The design result of the final structure was determined to be 1.54 mm. The laminate sequence is [45°/0°/45°/0°/45°/0°/45°]. Table 3 shows a summary of the structural analysis results for the air intake structure. Figure 13, Figure 14 and Figure 15 show the result of stress and displacement through the final structural analysis of load case one. Structural analysis was performed for a total of six cases in this study. The tensile stress and compressive stress were confirmed to be sufficiently safe as a result of examining the safety factors. It was confirmed that it was sufficiently safe when compared with the results of other studies on the design and analysis of aircraft to which the composite material was applied.

## 6. Conclusions

The aim of this work was to design the engine intake structure of a small aircraft. For structural safety evaluation, a finite element analysis method was applied. In this study, structural design and analysis were carried out on the s-duct and engine intake, which is an aircraft structure with composite materials applied, to examine the structural safety. The netting rule and the rule of mixture considering the composite laminate theory were used for the initial structural design. The target structure consists of two parts. It consists of a plate part and a curved panel. The design of plate part was performed considering the laminate constitutive theory. Structural analysis was carried out using MSC NASTRAN, which is a commercial finite element analysis software. For the load used for structural analysis, the safety factor of 1.5 was applied after considering the pressure distribution load and the acceleration condition, and the boundary condition was applied as the fixed boundary condition of the connecting part. The structural analysis examined stress, displacement, and buckling. As a result of examining the stress and displacement analysis results, it was determined to be a safe structure. As a result of examining the vulnerability to buckling, it was confirmed to be stable enough. Therefore, the structural design result through this study was analyzed to be valid. In the future, manufacturing of a prototype will be carried out using the structural design results presented in this work. Furthermore, future structural tests are planned to reflect the structural analysis results.

## Figures and Tables

**Figure 1 materials-15-03001-f001:**
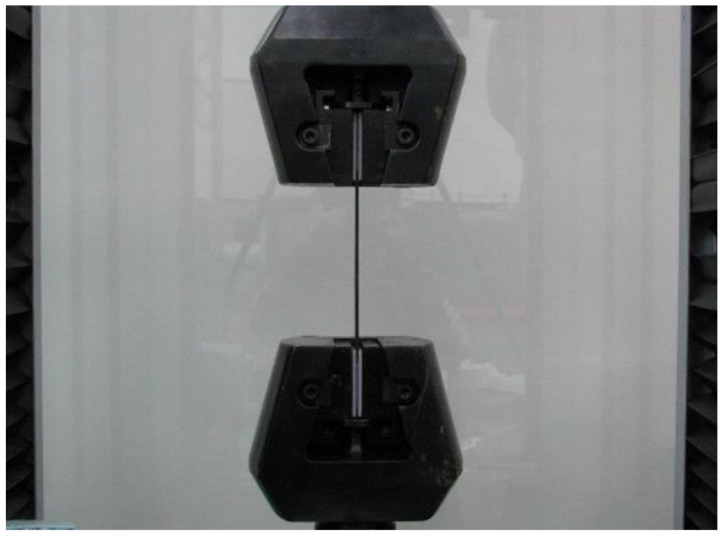
Tensile test of the manufactured specimen.

**Figure 2 materials-15-03001-f002:**
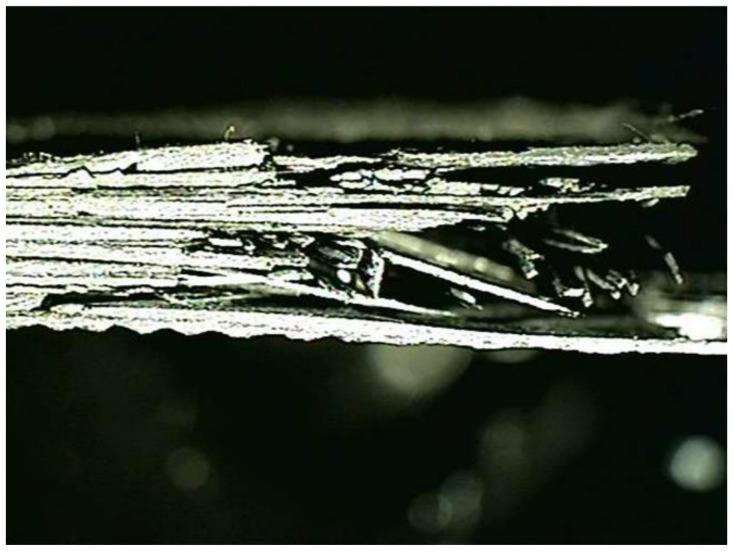
Investigation of the fracture surface of the tensile specimen.

**Figure 3 materials-15-03001-f003:**
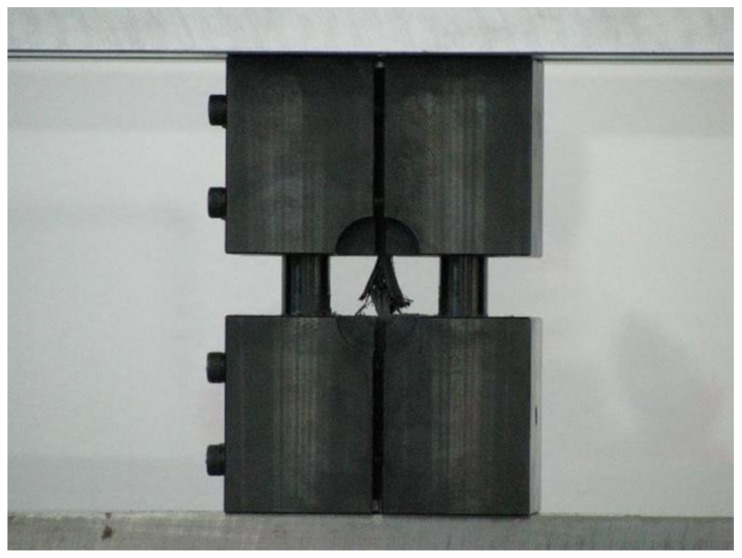
Compression test of the manufactured specimen.

**Figure 4 materials-15-03001-f004:**
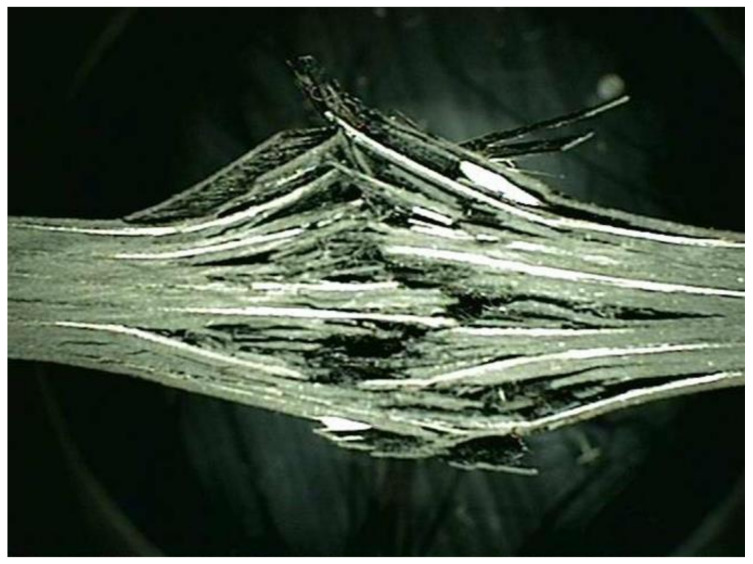
Investigation of the fracture surface of the compression specimen.

**Figure 5 materials-15-03001-f005:**
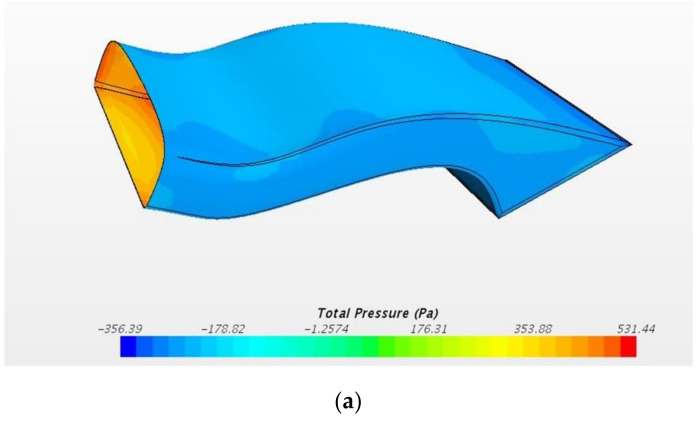
Pressure distribution by CFD analysis. (**a**) Pressure distribution of intercooler; (**b**) pressure distribution of radiator.

**Figure 6 materials-15-03001-f006:**
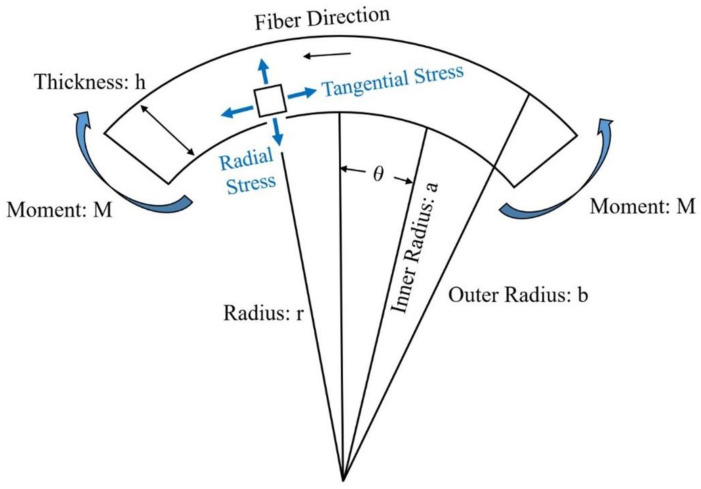
Curved beam configuration.

**Figure 7 materials-15-03001-f007:**
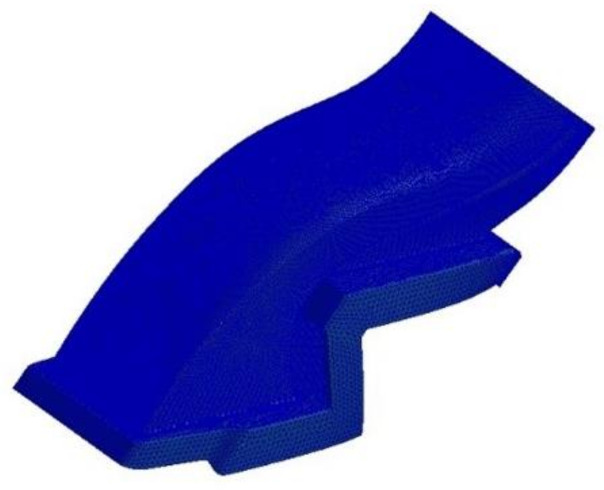
Finite element model of S-duct.

**Figure 8 materials-15-03001-f008:**
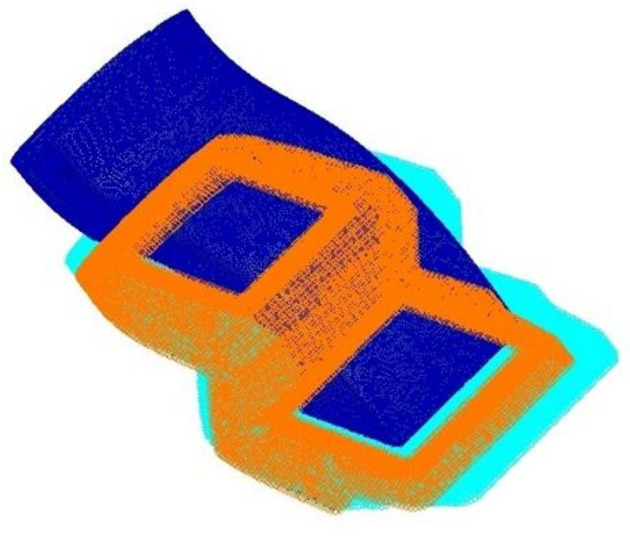
Boundary condition of S-duct.

**Figure 9 materials-15-03001-f009:**
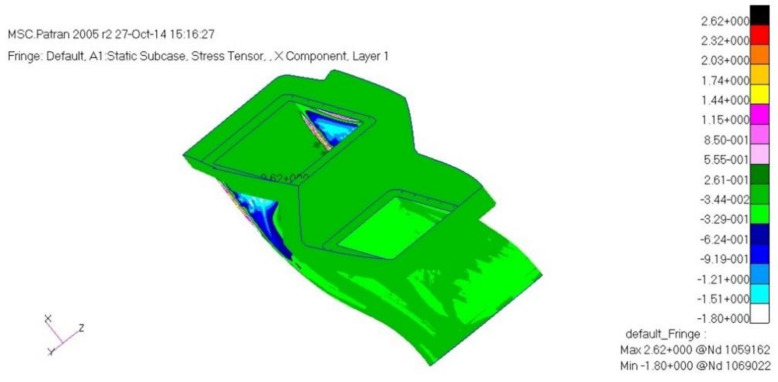
Stress analysis result: layer 1.

**Figure 10 materials-15-03001-f010:**
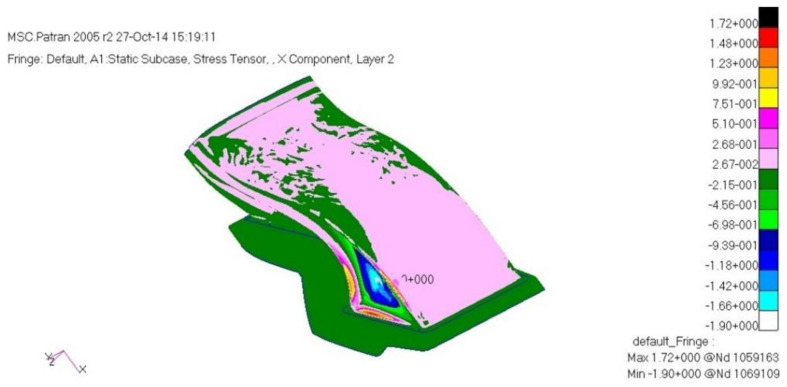
Stress analysis result: layer 2.

**Figure 11 materials-15-03001-f011:**
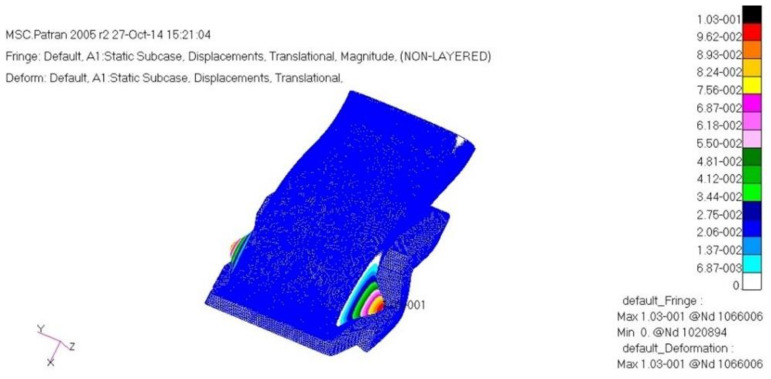
Displacement analysis result.

**Figure 12 materials-15-03001-f012:**
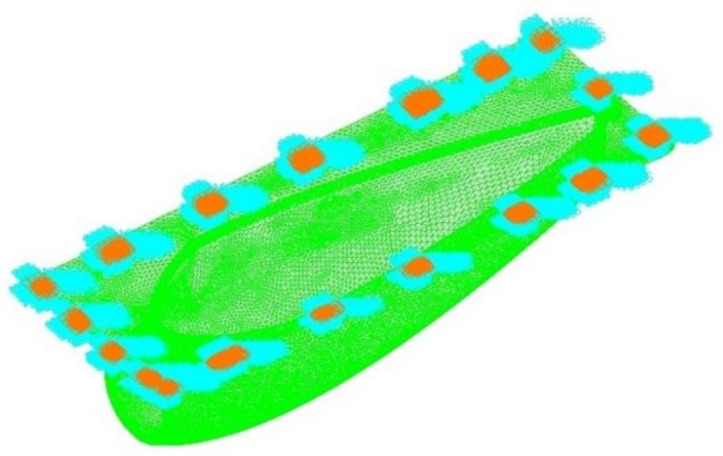
Boundary condition.

**Figure 13 materials-15-03001-f013:**
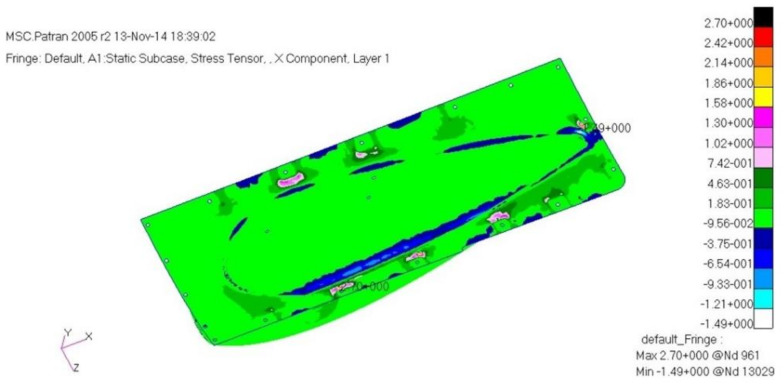
Stress analysis result: layer 1.

**Figure 14 materials-15-03001-f014:**
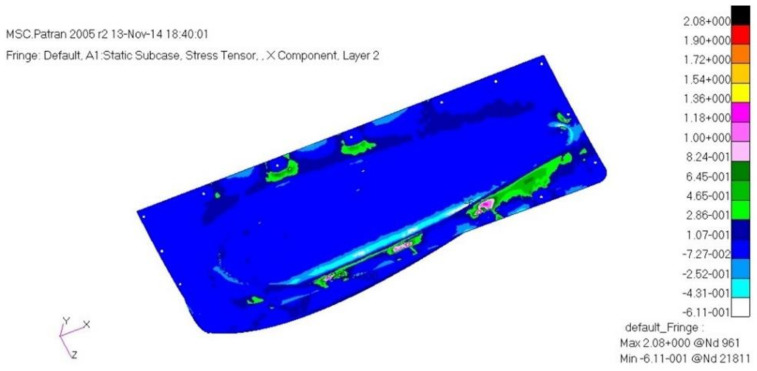
Stress analysis result: layer 2.

**Figure 15 materials-15-03001-f015:**
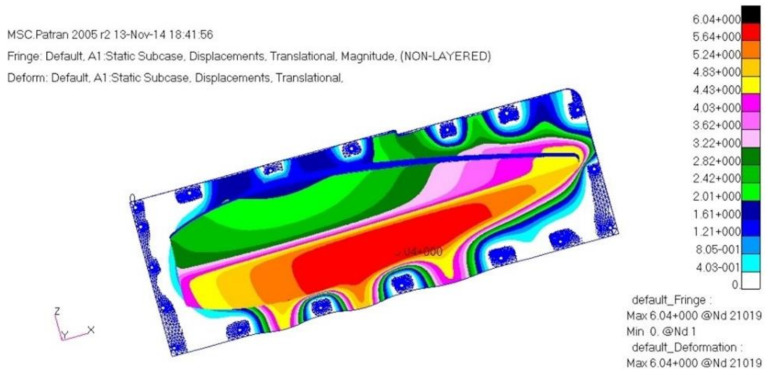
Displacement analysis result.

**Table 1 materials-15-03001-t001:** Load cases.

Load Cases	Acceleration	Pressure	
		Outside (Pa)	Inside (Pa)
Load case 1	*z* axis −6G	−424.63	725.5
Load case 2	*z* axis +3G	−424.63	725.5
Load case 3	*y* axis +2G	−424.63	725.5
Load case 4	*x* axis +20G	−424.63	725.5
Load case 5	*y* axis +20G	−424.63	725.5
Load case 6	*z* axis +20G	−424.63	725.5

**Table 2 materials-15-03001-t002:** Structural analysis results of s-duct structure.

Load Cases	Acceleration	Stress	
		Tension (MPa)	Compression (MPa)
Load case 1	*z* axis −6G	2.62	1.8
Load case 2	*z* axis +3G	2.62	1.8
Load case 3	*y* axis +2G	1.9	1.07
Load case 4	*x* axis +20G	5.28	2.36
Load case 5	*y* axis +20G	1.9	1.07
Load case 6	*z* axis +20G	2.62	1.8

**Table 3 materials-15-03001-t003:** Structural analysis results of air intake structure.

Load Cases	Acceleration	Stress	
		Tension (MPa)	Compression (MPa)
Load case 1	*z* axis −6G	2.7	1.49
Load case 2	*z* axis +3G	2.7	1.49
Load case 3	*y* axis +2G	2.11	1.13
Load case 4	*x* axis +20G	24.8	12.4
Load case 5	*y* axis +20G	2.11	1.13
Load case 6	*z* axis +20G	2.7	1.49

## Data Availability

The data presented in this study are available on reasonable request from the corresponding author.

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
