# Peer review of "A Study on Design of S-Duct Structures and Air Intake for Small Aircraft Applied to High Strength Carbon–Epoxy Composite Materials"

_materials, 2022, doi:10.3390/ma15093001_

Round 1

Reviewer 1 Report

REVIEW

on article

A Study on Design of S-Duct Structures and Air Intake for Small Aircraft Applied to High Strength Carbon-Epoxy Composite Materials

 Semyeong Lim, Won Choi and Hyunbum Park

SUMMARY

The article is devoted to solving urgent scientific and practical problems. The aim of the work was to study the design of S-Duct structures and air intake for small aircraft using high-strength carbon-epoxy composite materials.

The authors performed a large-scale comprehensive study. They carried out structural design and numerical analysis of the air intake and s-shaped duct structures for a small aircraft. A study was made of the mechanical properties of the carbon/epoxy material. When designing the structure, the authors used the conditions of a distributed load of pressure and acceleration.

The authors paid attention to security. The design load was analyzed taking into account the safety factor.

The results of the authors are original and verified by them in the course of research. To analyze the validity of the design calculation results, a structural analysis was carried out.

The article is practically significant for the applied industry. Structural analysis using the finite element method confirmed that the designed air intake structure was safe. Based on the design results, a design prototype was manufactured.

The article makes a positive impression, but needs some improvement. Below are the comments of the reviewer.

COMMENTS

  1. The Abstract must be structured and expanded according to the plan: Background, Purpose, Method, main results and Conclusions. The Abstract presented by the authors contains only a listing of what has been done.
  2. The authors conducted a rather concise literature review on the topic of the study, in the "Introduction" section there are only 10 references to the works of other authors. It is necessary to expand the literature review mainly with recent research in the presented field, adding 10-15 literature sources.
  3. As a scientific research problem, the authors very succinctly note that little research has been done on the design and analysis of the UAV composite structure. This formulation needs some expansion and addition. It should be indicated in 2-3 sentences why this scientific deficit should be filled in the study.
  4. At the end of the "Introduction" section, the statement of the purpose and objectives of the study should be added.
  5. According to the text, there are differences in the spelling of Figure and Fig. It is necessary to bring to the correct uniformity.
  6. It is necessary to justify in more detail the choice of the material used in the study – a high-strength carbon-epoxy composite. You should add 2-3 theses in favor of your choice.
  7. Section 2 ends with four figures and one table without any comments on them. When mentioning figures and their analysis, only references [15] and [16, 17] are given. After the figures and the table, it is necessary to add several paragraphs in which to analyze the form of destruction of the samples and the obtained mechanical properties of the carbon/epoxy material.
  8. The test equipment used (type, manufacturer, country) should be described in more detail or referenced.
  9. After formulas 1-4, it is necessary to add explanations to them, that is, describe all the parameters given.
  10. The section "Conclusion" contains a listing of what the authors have done and a lot of information about the research method. It is necessary to highlight the results obtained in the study, their practical significance. It is advisable to paint it point by point. The Conclusion need to be reworked.
  11. Links in the form https://doi.org/ or a doi number should be added to References to make it easier to find works cited by authors.

Author Response

Title: A Study on Design of S-Duct Structures and Air Intake for Small Aircraft Applied to High Strength Carbon-Epoxy Composite Materials

Authors: Semyeong Lim, Won Choi, Hyunbum Park

Dear editor,

Thank you for your useful comments of the manuscript. I have modified the manuscript accordingly, and the detailed corrections are listed below point by point:

<Reviewer#1>

  1. The Abstract must be structured and expanded according to the plan: Background, Purpose, Method, main results and Conclusions. The Abstract presented by the authors contains only a listing of what has been done.

→ We revised the manuscript following reviewer’s comment. The abstract has been restructured and rewritten.

Recently, many structural parts using composite materials are being applied to small aircraft and UAV in the world. The aim of this work is to design the engine intake structure of a small aircraft. For structural safety evaluation, a finite element analysis method was applied. In this work, structural design and numerical analysis of air intake and s-duct structures for small aircraft was performed. The target structure is composed of an s-duct and a cylindrical intake structure. Firstly, investigation on mechanical properties of carbon/epoxy material was conducted. The distributed pressure load and acceleration condition was applied to structural design. The design load was analyzed in consideration of the safety factor. The structural analysis was performed to analyze the validity of the structural design results. Through the structural analysis using finite element method, it was confirmed that the designed air intake structure is safety. Based on the designed result, the manufacturing of prototype structure was carried out.

  1. The authors conducted a rather concise literature review on the topic of the study, in the "Introduction" section there are only 10 references to the works of other authors. It is necessary to expand the literature review mainly with recent research in the presented field, adding 10-15 literature sources.

→ We revised the manuscript following reviewer’s comment. The literature reviews were added in the manuscript.

[11] G. Song, J. Li, M. Tang, Y. Wu, Y. Luo. Experimental simulation methodology and spatial transition of complex distortion fields in a S-shaped inlet. Aerospace Science and Technology 2021, 116, 106855, doi:10.1016/j.ast.2021.106855.

[12] M. Wojeodka, C. White, S. Shahpar, K. Kontis. Numerical study of complex flow physics and coherent structures of the flow through a convoluted duct. Aerospace Science and Technology 2022, 121, 107191, doi:10.1016/j.ast.2021.107191.

[13] L. Jun, Y. Huacheng, C. Wenfang, G. Rongwei. Investigation of translation scheme turbine-based combined-cycle inlet mode transition. Aerospace Science and Technology 2021, 116, 106894, doi:10.1016/j.ast.2021.106894.

[14] G. Kim, B. Lee, H. Lu, J. Park. Rongwei. Failure analysis of an aircraft APU exhaust duct flange due to low cycle fatigue at high temperatures. Engineering Failure Analysis 2012, 20, 97-104, doi:10.1016/j.engfailanal.2011.11.003.

[15] M. Wroblewski, M. Adamczyk, A. Kozakiewicz. Areas of investigation into air intake systems for the impact on compressor performance stability in aircraft turbine engines. Adv. Sci. Technol. Res. J. 2022, 16, 62-74, doi:10.12913/22998624/143290.

[16] A. Kozakiewicz, M. Frant, M. Majcher. Impact of the intake vortex on the stability of the turbine jet engine intake system. International Review of Aerospace Engineering. 2021, 14, 173-180, doi:10.15866/irease.v14i4.20223.

[17] A. Kozakiewicz, M. Frant. Analysis of the gust impact on inlet vortex formation of the fuselage-shielded inlet of an jet engine powered aircraft. Journal of Theoretical and Applied Mechanics. 2013, 51, 993-1002.

[18] C. Soutis. Fibre reinforced composites in aircraft construction. Progress in Aerospace Sciences 2005, 41, 143-151, doi:10.1016/j.paerosci.2005.02.004.

[19] A. Grobovic, G. Kastratovic, Z. Bozic, Z. Bozic, A. Obradovic, A. Sedmak, S. Sedmak. Experimental and numerical evaluation of fracture characteristics of composite material used in the aircraft engine cover manufacturing. Engineering Failure Analysis 2022, 137, 106286, doi:10.1016/j.engfailanal.2022.106286.

[20] M. Finley. Composites make for greener aircraft engines. Reinforced Plastics 2008, 52, 24-25, doi:10.1016/S0034-3617(08)70033-X.

  1. As a scientific research problem, the authors very succinctly note that little research has been done on the design and analysis of the UAV composite structure. This formulation needs some expansion and addition. It should be indicated in 2-3 sentences why this scientific deficit should be filled in the study.

→ We revised the manuscript following reviewer’s comment. The direction of this study was extended to the field of small aircraft composite structures. Therefore, the text in the manuscript has been modified.

Many research works of aerodynamic design for air intake of aircraft were per-formed. However, little research work has been carried out to design and numerical analysis of small aircraft structure using composite materials.

  1. At the end of the "Introduction" section, the statement of the purpose and objectives of the study should be added.

→ We revised the manuscript following reviewer’s comment.

The aim of this work is to design the engine intake structure of a small aircraft. In this study, high strength carbon/epoxy composite was applied to conduct the structural de-sign and analysis of small aircraft structure. For structural safety evaluation, a finite element analysis method was applied. Finally, structural design and numerical analysis of air intake and s-duct structures for small aircraft was performed.

  1. According to the text, there are differences in the spelling of Figure and Fig. It is necessary to bring to the correct uniformity.

→ We revised the manuscript following reviewer’s comment. The manuscript was revised to reflect “Figure” character.

  1. It is necessary to justify in more detail the choice of the material used in the study – a high-strength carbon-epoxy composite. You should add 2-3 theses in favor of your choice.

→ We revised the manuscript following reviewer’s comment.

In this study, high strength carbon/epoxy composite was adopted for structural design. The carbon fiber and epoxy matrix is a material optimized for aircraft structures to reduce weight. Recently, most aircraft are designed by adopting carbon fiber for weight reduction. Therefore, carbon fiber was adopted for the structural design of the target small aircraft.

  1. Section 2 ends with four figures and one table without any comments on them. When mentioning figures and their analysis, only references [15] and [16, 17] are given. After the figures and the table, it is necessary to add several paragraphs in which to analyze the form of destruction of the samples and the obtained mechanical properties of the carbon/epoxy material.

→ We revised the manuscript following reviewer’s comment.

In the case of tensile specimens, the fiber fracture behavior after matrix damage is shown. In the case of compression specimens, the fibers at both ends are broken by compression, showing the type of damage.

  1. The test equipment used (type, manufacturer, country) should be described in more detail or referenced.

→ We revised the manuscript following reviewer’s comment.

The Universal Testing Machine of Shimadzu AG-250KNX was applied. The equipment was manufactured by SUNGSAH HI TECH in Rep. of Korea. The test speed of equipment is 0.001~500mm/min.

  1. After formulas 1-4, it is necessary to add explanations to them, that is, describe all the parameters given.

→ We revised the manuscript following reviewer’s comment.

The following equations were used to design the curved structure of the duct. σr is radial stress, k is shear coefficients. σθ is tangential stress. The structural design was performed by comparing the stress calculated using the following equation and the yield strength of the material under the applied load.

  • The section "Conclusion" contains a listing of what the authors have done and a lot of information about the research method. It is necessary to highlight the results obtained in the study, their practical significance. It is advisable to paint it point by point. The Conclusion need to be reworked.

→ We revised the manuscript following reviewer’s comment. The Conclusion was revised.

The aim of this work is to design the engine intake structure of a small aircraft. For structural safety evaluation, a finite element analysis method was applied. In this study, structural design and analysis was carried out on the s-duct and engine intake, which is an aircraft structure with composite materials applied, to examine the structural safety. The netting rule and the rule of mixture considering on composite laminate theory were used for initial structural design. The target structure consists of two parts. It consists of a plate part and a curved panel. The design of plate part was per-formed considering on laminate constitutive theory. Structural analysis was carried out by using MSC. NASTRAN which is a commercial finite element analysis software. For the load for structural analysis, the safety factor of 1.5 was applied after considering the pressure distribution load and the acceleration condition, and the boundary condition was applied as the fixed boundary condition of the connecting part. For the structural analysis, it was carried out three analyses such as stress, displacement, and buckling analysis. As a result of examining the stress and displacement analysis results, it was examined to be a safe structure. As a result of examining the vulnerability to buckling, it was confirmed to be stable enough. Therefore, the structural design result through this study was analyzed to be valid. In the future, manufacturing of prototype will be carried out using the structural design results presented in this work. Structural tests are planned to reflect the structural analysis results.

  • Links in the form https://doi.org/ or a doi number should be added to References to make it easier to find works cited by authors.

→ We revised the manuscript following reviewer’s comment. The doi number were added in the manuscript.

April 5. 2022.

Hyunbum Park

Reviewer 2 Report

The purpose of the manuscript is to design S-Duct structures and air intakes for small aircraft applied to high-strength carbon-epoxy composite materials. The study suggests that structural analysis based on the FE method confirms that the designed air intake structure is safe. In this context, based on the projected result, it was possible to manufacture the prototype structure. The results are explained in detail.

The research is an original one whose contribution is felt by producing new knowledge based on the approach of the debated topic.

The submitted manuscript on ‘A Study on Design of S-Duct Structures and Air Intake for Small Aircraft Applied to High Strength Carbon-Epoxy Composite Materials’ is well written and interesting to read, however I see some minor issues that should be resolved before publishing this paper.

The manuscript highlights good insights on the topic and adds some value to the literature.

The abstract section looks good. The authors considered the background, materials & methods, results & discussion, conclusion when writing the abstract of an original article.

The introduction section looks good and well written by the authors. But I would like to suggest highlighting the paper purpose and to present the knowledge gap of the research.

The manuscript ends with the conclusion providing a summary of the activities undertaken in this work.  In the Conclusion section I would like to suggest that the authors add a paragraph highlighting the output responses of the experimental work.

I suggest making the paper's contribution be clearer in the introduction and conclusion.

Present the future scope of research.

Author Response

Title: A Study on Design of S-Duct Structures and Air Intake for Small Aircraft Applied to High Strength Carbon-Epoxy Composite Materials

Authors: Semyeong Lim, Won Choi, Hyunbum Park

Dear editor,

Thank you for your useful comments of the manuscript. I have modified the manuscript accordingly, and the detailed corrections are listed below point by point:

<Reviewer#2>

The purpose of the manuscript is to design S-Duct structures and air intakes for small aircraft applied to high-strength carbon-epoxy composite materials. The study suggests that structural analysis based on the FE method confirms that the designed air intake structure is safe. In this context, based on the projected result, it was possible to manufacture the prototype structure. The results are explained in detail.

The research is an original one whose contribution is felt by producing new knowledge based on the approach of the debated topic.

The submitted manuscript on ‘A Study on Design of S-Duct Structures and Air Intake for Small Aircraft Applied to High Strength Carbon-Epoxy Composite Materials’ is well written and interesting to read, however I see some minor issues that should be resolved before publishing this paper.

→ We revised the manuscript following reviewer’s comment.

The manuscript highlights good insights on the topic and adds some value to the literature.

→ The literature reviews were added another reviewer’s comment.

[11] G. Song, J. Li, M. Tang, Y. Wu, Y. Luo. Experimental simulation methodology and spatial transition of complex distortion fields in a S-shaped inlet. Aerospace Science and Technology 2021, 116, 106855, doi:10.1016/j.ast.2021.106855.

[12] M. Wojeodka, C. White, S. Shahpar, K. Kontis. Numerical study of complex flow physics and coherent structures of the flow through a convoluted duct. Aerospace Science and Technology 2022, 121, 107191, doi:10.1016/j.ast.2021.107191.

[13] L. Jun, Y. Huacheng, C. Wenfang, G. Rongwei. Investigation of translation scheme turbine-based combined-cycle inlet mode transition. Aerospace Science and Technology 2021, 116, 106894, doi:10.1016/j.ast.2021.106894.

[14] G. Kim, B. Lee, H. Lu, J. Park. Rongwei. Failure analysis of an aircraft APU exhaust duct flange due to low cycle fatigue at high temperatures. Engineering Failure Analysis 2012, 20, 97-104, doi:10.1016/j.engfailanal.2011.11.003.

[15] M. Wroblewski, M. Adamczyk, A. Kozakiewicz. Areas of investigation into air intake systems for the impact on compressor performance stability in aircraft turbine engines. Adv. Sci. Technol. Res. J. 2022, 16, 62-74, doi:10.12913/22998624/143290.

[16] A. Kozakiewicz, M. Frant, M. Majcher. Impact of the intake vortex on the stability of the turbine jet engine intake system. International Review of Aerospace Engineering. 2021, 14, 173-180, doi:10.15866/irease.v14i4.20223.

[17] A. Kozakiewicz, M. Frant. Analysis of the gust impact on inlet vortex formation of the fuselage-shielded inlet of an jet engine powered aircraft. Journal of Theoretical and Applied Mechanics. 2013, 51, 993-1002.

[18] C. Soutis. Fibre reinforced composites in aircraft construction. Progress in Aerospace Sciences 2005, 41, 143-151, doi:10.1016/j.paerosci.2005.02.004.

[19] A. Grobovic, G. Kastratovic, Z. Bozic, Z. Bozic, A. Obradovic, A. Sedmak, S. Sedmak. Experimental and numerical evaluation of fracture characteristics of composite material used in the aircraft engine cover manufacturing. Engineering Failure Analysis 2022, 137, 106286, doi:10.1016/j.engfailanal.2022.106286.

[20] M. Finley. Composites make for greener aircraft engines. Reinforced Plastics 2008, 52, 24-25, doi:10.1016/S0034-3617(08)70033-X.

The abstract section looks good. The authors considered the background, materials & methods, results & discussion, conclusion when writing the abstract of an original article.

The introduction section looks good and well written by the authors. But I would like to suggest highlighting the paper purpose and to present the knowledge gap of the research.

→ We revised the manuscript following reviewer’s comment. The introduction has been restructured and rewritten to reflect highlighting the paper purpose.

The manuscript ends with the conclusion providing a summary of the activities undertaken in this work. In the Conclusion section I would like to suggest that the authors add a paragraph highlighting the output responses of the experimental work.

→ We revised the manuscript following reviewer’s comment.

The aim of this work is to design the engine intake structure of a small aircraft. For structural safety evaluation, a finite element analysis method was applied. In this study, structural design and analysis was carried out on the s-duct and engine intake, which is an aircraft structure with composite materials applied, to examine the structural safety. The netting rule and the rule of mixture considering on composite laminate theory were used for initial structural design. The target structure consists of two parts. It consists of a plate part and a curved panel. The design of plate part was per-formed considering on laminate constitutive theory. Structural analysis was carried out by using MSC. NASTRAN which is a commercial finite element analysis software. For the load for structural analysis, the safety factor of 1.5 was applied after considering the pressure distribution load and the acceleration condition, and the boundary condition was applied as the fixed boundary condition of the connecting part. For the structural analysis, it was carried out three analyses such as stress, displacement, and buckling analysis. As a result of examining the stress and displacement analysis results, it was examined to be a safe structure. As a result of examining the vulnerability to buckling, it was confirmed to be stable enough. Therefore, the structural design result through this study was analyzed to be valid. In the future, manufacturing of prototype will be carried out using the structural design results presented in this work. Structural tests are planned to reflect the structural analysis results.

I suggest making the paper's contribution be clearer in the introduction and conclusion.

Present the future scope of research.

→ We revised the manuscript following reviewer’s comment.

In the future, manufacturing of prototype will be carried out using the structural de-sign results presented in this work. Structural tests are planned to reflect the structural analysis results.

April 5. 2022.

Hyunbum Park

Round 2

Reviewer 1 Report

The authors significantly revised the article. Now it looks much better.

Several remarks:

  1. The article lacks the Discussion section. I recommend add the Discussion section or rename 5th section "Structural Analysis of Air intake and results Discussion", and give several paragraphs where the authors compare their results with data of other researchers.
  2. Please, check all Figures for quality: 1000 pix for the smallest side and 300 DPI resolution.

Author Response

  1. The article lacks the Discussion section. I recommend add the Discussion section or rename 5th section "Structural Analysis of Air intake and results Discussion", and give several paragraphs where the authors compare their results with data of other researchers.

→ We revised the manuscript following reviewer’s comment.

Structural analysis was performed for a total of 6 cases in this study. The tensile stress and compressive stress were confirmed to be sufficiently safe as a result of examining the safety factor. It was confirmed that it was sufficiently safe when compared with the results of other studies, the design and analysis of aircraft to which the composite material was applied.

  1. Please, check all Figures for quality: 1000 pix for the smallest side and 300 DPI resolution.

→ We revised the manuscript following reviewer’s comment. The quality of Figures was revised.
